# Identifying Patients in Whom the Follow-Up Scheme after Robot-Assisted Radical Prostatectomy Could Be Optimized in the First Year after Surgery: Reducing Healthcare Burden

**DOI:** 10.3390/biomedicines11030727

**Published:** 2023-02-28

**Authors:** Hans Veerman, Sophia H. van der Graaf, Dennie Meijer, Marinus J. Hagens, Corinne N. Tillier, Pim J. van Leeuwen, Henk G. van der Poel, André N. Vis

**Affiliations:** 1Department of Urology, Netherlands Cancer Institute, Antoni van Leeuwenhoek Hospital, 1066 CX Amsterdam, The Netherlands; 2Department of Urology, Amsterdam University Medical Centers, 1066 CX Amsterdam, The Netherlands; 3Prostate Cancer Network the Netherlands, 1066 CX Amsterdam, The Netherlands

**Keywords:** prostatic neoplasms, prostatectomy, prostate-specific antigen, follow-up, aftercare, recurrence, delivery of healthcare

## Abstract

Background: The currently advised follow-up scheme of PSA testing after robot-assisted radical prostatectomy (RARP) is strict and might pose a burden to our healthcare system. We aimed to optimize the 1-year follow-up scheme for patients who undergo RARP. Methods: All patients with histologically-proven prostate cancer (PCa) who underwent RARP between 2018 and August 2022 in the Prostate Cancer Network in the Netherlands were retrospectively evaluated. We excluded patients who underwent salvage RARP and patients who had <1 year of PSA follow-up. Postoperative PSA values were collected. Biochemical persistence (BCP) was defined as PSA level >0.10 ng/mL at 0–4 months after RARP, whereas biochemical recurrence (BCR) was defined as PSA level >0.2 ng/mL at any time point after RARP. We aimed to identify a group of patients who had a very low risk of BCR at different time points after surgery. Results: Of all 1155 patients, BCP was observed in 151 (13%), of whom 79 (6.8%) had PSA ≥ 0.2 ng/mL. BCR further developed in 51 (4.7%) and 37 (3.4%) patients at 5–8 and 9–12 months after RARP, respectively. In 12 patients, BCR was found at 5–8 months after RARP in the absence of BCP. These patients represented 1.2% (12/1004) of the entire group. In other words, 98.8% (992/1004) of patients who had an unmeasurable PSA level at 0–4 months after RARP also had an unmeasurable PSA level 5–8 months after surgery. Limitations are the retrospective design and incomplete follow-up. Conclusions: Patients with an unmeasurable PSA level at 3–4 months after RARP may not need to be retested until 12 months of follow-up, as almost 100% of patients will not have the biochemically recurrent disease at 5–8 months of follow-up. This will reduce PSA testing substantially at the cost of hardly any missed patients with recurrent disease.

## 1. Introduction

Prostate Cancer (PCa) is the second most common malignancy in men in the western world [1]. It is estimated that 1 in 9 men will develop PCa, which is associated with a high workload for the urologist [2]. Localized PCa can be managed with several treatment modalities, i.e., active surveillance, brachytherapy, external beam radiotherapy, and (robot-assisted) radical prostatectomy (RARP). For each treatment modality, follow-up is performed through prostate-specific antigen (PSA) monitoring. For instance, the European Association of Urology (EAU) guidelines recommend the determination of serum-PSA level every three to four months during the first year after surgery, every six months in the second year after surgery, and annually thereafter [3]. It is obvious that this follow-up scheme is very strict and is not personalized to an individual patient or tumor characteristics. For patients, this means visiting the hospital several times per year for blood draw. For healthcare professionals, this means contacting the patient several times to discuss the blood test results. It goes without saying that the presented follow-up scheme after RARP is not very purposeful, as not every patient has the same risk of recurrence after RARP [4]. Moreover, if screening regimens are adapted in the future, follow-up schemes in those after curative treatments need to be more optimized [5].

To assess whether the follow-up scheme of patients undergoing RARP could be optimized in the first year after surgery, we studied the biochemical recurrence (BCR) rate after surgery at each time-point after surgery in a large series of patients who underwent RARP in the Prostate Cancer Network the Netherlands (PCNN).

## 2. Patients and Methods

A multicenter retrospective study was performed at the RARP centers of the Prostate Cancer Network Netherlands (PCNN). The PCNN is a collaboration of 12 Dutch hospitals aimed to improve the quality of PCa diagnosis and management. The study was approved by the local institutional review board (IRBd19-248). The need for informed consent was waived.

### 2.1. Patients

We retrospectively analyzed a prospectively maintained database of histologically-proven PCa patients who underwent RARP at the PCNN between January 2018 and August 2022. Patients who underwent salvage RARP were excluded (N = 8) (Figure 1). Patients who had less than 1 year of PSA follow-up were also excluded (N = 962).

### 2.2. PSA Follow-Up

Patients received follow-ups in different centers within the PCNN. PSA follow-up was performed at the outpatient clinic via slightly different follow-up schedules and depended somewhat on the hospital performing the follow-up after RARP. PSA was measured at either 3 months, 6 months, 9 months, and 12 months in the first year after surgery, every 6 months in the second year and annually thereafter, or at 4 months, 8 months, and 12 months in the first year after RARP, every 6 months in the second year and annually thereafter. The use of conventional PSA tests (detectable from 0.1 ng/mL) and ultrasensitive PSA tests (detectable from 0.01 ng/mL) varied between centers. Referring centers were contacted to provide PSA results.

### 2.3. Data and Outcomes

The following variables were collected: age (years), pathological tumor stage (pT-stage), pathological nodal stage (pN-stage), pathological international society of urological pathologists (ISUP) grade group, the presence of a positive surgical margin (PSM), and postoperative PSA-values. Biochemically persistent PSA (BCP) was defined as any PSA >0.10 ng/mL at first measurement at 3 or 4 months of follow-up after RARP. BCR was defined as a PSA >0.20 ng/mL 0–4 months of follow-up or at later time points after RARP.

### 2.4. Statistical Analysis

Continuous variables were reported as medians with inter-quartile range (IQR). Nominal variables were reported as frequencies and percentages (%). Given the longitudinal nature of the timing of BCR, we recoded patients in three different timeframes of recurrence within 1 year to: (1) BCR 0–4 months after RARP, (2) 5–8 months after RARP, and (3) 8–12 months after RARP. Logistic regression analysis was applied to find risk factors for BCR within one year after RARP in the subgroup of patients with unmeasurable PSA 0–4 months after RARP. As a rule of thumb, only 1 variable per 10 events was added to the multivariable logistic regression analysis. With these risk factors, low- and high-risk groups for BCR were identified. Kaplan–Meier curves and the Log-rank test were used to depict and analyze the difference in BCR-free survival between the risk groups. Statistical significance was set at *p* < 0.05.

## 3. Results

In total, 1155 patients who underwent RARP at the PCNN were analyzed. The baseline characteristics of all patients were reported in Table 1. BCP was observed in 151 (13%) patients, of whom 79 (6.8%) had PSA > 0.2 ng/mL. In total, 169 (14.6%) patients had BCR < 1 year after RARP. BCR developed in 79 (6.8%), 51 (4.7%), and 37 (3.4%) patients at 0–4, 5–8, and 9–12 months after RARP, respectively (Table 2). The median PSA follow-up in 856 patients without BCR was 25 months (IQR 19–33).

### 3.1. Risk Factors for BCR < 1 Year after RARP with Unmea Surable PSA

Of 1004 patients with unmeasurable PSA-level at the first time-point after RARP, 12 (1.2%) and 24 (2.4%) patients experienced BCR at 5–8 months and 9–12 months after RARP. Risk factors for developing BCR <1 year after RARP without BCP (in univariable logistic regression analysis) were pT3a, pT3b, pN1, ISUP 5, and PSM (Table 3). In multivariable analysis, pN1, ISUP 5, pT3b, and PSM were significant risk factors for relapse.

When the subgroup of patients was analyzed who had no BCP, pathological ISUP grade 1–4, and absence of lymph-node metastases (pNx/pN0) (low-risk group), only 16/833 (1.9%) experienced BCR < 1 year after RARP. In the subgroup of patients without BCP and with pathological ISUP grade 5 or pN1 (high-risk group), the frequency of BCR after RARP increased to 18/171 (11%). The BCR-free survival over time was significantly different between the low-risk group for BCR and the high-risk group for BCR (*p* < 0.001) (Figure 2a,b).

### 3.2. In Whom Can PSA Testing at 5–8 Months after RARP Be Omitted Safely?

The group of patients without BCP who developed BCR < 1 year after RARP consisted of 34 patients. In 12/34 patients, BCR was found after 5–8 months; this represents 1.2% (12/1004) of the entire group without BCP. In other words, 98.8% (992/1004) of patients with unmeasurable PSA had a normal PSA 5–8 months after RARP. This proportion increased to 99.3% in the subgroup of patients without BCP who had both pathological ISUP grade 1–4 and pNx/pN0. Out of 171 patients without BCP and with pN1 or pathological ISUP grade 5 tumors, 6 (3.5%) developed BCR 5–8 months after RARP.

Out of 12 patients with a BCR at 5–8 months, the exact PSA values were unavailable for two patients, and two other patients underwent salvage radiotherapy before the subsequent PSA test. For the remaining 8 patients with PSA tested at both 8 and 12 months after RARP, the PSA value was still in the optimal range (0.2–0.5 ng/mL) for salvage treatment 1 year after surgery in 6 patients (75%) [6,7,8,9]. Additionally, all patients met the criteria of EAU high-risk for BCR (PSA-doubling time < 1 year or pathological ISUP grade 4–5) which also justifies further salvage treatment [10].

### 3.3. Reducing the Burden

Omitting the PSA test 5–8 months after RARP for all patients with unmeasurable PSA would result in 93% fewer PSA tests (1004 out of 1076 patients who had no BCR 0–4 months after RARP) while missing 1.2% (12/1004) of BCR. Omitting PSA test 5–8 months after RARP only in the low-risk patients (pathological ISUP 1–4 and pN0/pNx) would result in 77% (833/1076) less PSA tests while missing BCR in 0.7% (6/833).

## 4. Discussion

The current study evaluated whether the follow-up scheme of patients undergoing RARP for locally-confined PCa could be optimized. This is important as the burden on our healthcare system rises due to increased screening efforts for prostate cancer and the aging of the male population. This is in part reflected by a rise in the number of men undergoing radical curative surgery for locally confined PCa in recent years.

We found that in patients who had an unmeasurable PSA level at first measurement at 0–4 months after surgery, biochemical recurrence of disease at later time points was observed only infrequently. In those with unmeasurable PSA level at 0–4 months after surgery, only 1.2% of patients experienced a measurable PSA level (PSA ≥ 0.2 ng/mL) at 5–8 months after RARP. So, the absolute number and relative proportion of PSA tests could be fairly reduced by over 90% at the downfall of missing 1.2% of patients with biochemically recurrent disease. We showed that in the absence of BCP, risk factors for biochemical recurrence (BCR) within 1 year after RARP were a positive lymph node dissection (pN1), pathological ISUP grade 5, seminal vesicle invasion (pT3b), and a positive surgical margin (PSM). If only patients with an unmeasurable PSA level at the first measurement, with ISUP 1–4 and no lymph node metastases (pN0/pNx) were considered, 77% of PSA tests could be omitted at 5–8 months after surgery at the downfall of missing BCR in 0.7% of patients.

Recurrent PSA after RARP is almost always followed by radiological and/or clinical recurrence [11]. Biochemically persistent PSA occurs in 5–20% of patients after radical prostatectomy (RP) and is associated with a worse prognosis, likely due to the presence of occult metastatic disease at the time of surgery [12,13]. A recent systematic review showed that BCR is a risk factor for developing distant metastases (DM), cancer-specific mortality (CSM), and overall mortality (OM) [14]. Additionally, a shorter time to recurrent PSA is associated with a higher risk of DM, CSM, and OS [14]. Therefore, it is crucial to identify patients with (early) BCR to apply salvage treatment before PSA exceeds 0.5 ng/mL [6,7,8,9]. On the other hand, the evidence for a specific follow-up schedule is low, and there are no prospective trials available evaluating the optimal timing of PSA testing after surgery [15]. Additionally, the current follow-up schedule proposed by the EAU and by other urological associations may be a little specific and probably not beneficial to a large proportion of patients.

Previous studies have tried to identify patients who might not benefit from rigorous PSA follow-up after radical prostatectomy. In a retrospective study, Swanson et al. found that only patients with ISUP > 3, an initial PSA > 20 ng/mL, or those with ISUP 2–3, as well as those with an initial PSA 10–20 ng/mL and pT3a/PSM, might benefit from rigorous and continuous PSA follow-up. The study group showed that the risk of CSM beyond 10 years after RP was significantly higher in patients with these tumor features than in those who did not have these tumor characteristics (22% vs. 4%) [16]. They suggested that patients with low-risk features may be discharged after 10 years of follow-up. Patients with pN1- or pT3b-status were excluded in the study of Swanson et al. Other studies showed that patients with pN1 or pT3b had an even higher risk of CSM and should definitely be monitored [17]. Matsumoto et al. analyzed the optimal follow-up scheme in patients after RARP based on the PSA-doubling time with the aim to find every single BCR before it exceeds 0.4 ng/mL [18,19]. Their optimal schedule was, for the most part, in line with the EAU guidelines, every 3 months for the first year, every 4 months for the second year, and semiannually thereafter [3]. The authors suggested that patients with measurable PSA values at baseline after RP (BCP; 0.1–0.199 ng/mL) might even need a more extensive follow-up scheme. The aim behind their optimal follow-up scheme was not to miss any BCR. With this in perspective, it may be assumed that for some patients, this follow-up scheme may be too strict.

We demonstrated that the absolute number and relative proportion of patients who need a PSA test 5–8 months after RARP could be fairly (and safely) reduced. If this follow-up scheme was to be introduced, the majority of patients do not need to visit the hospital for a blood draw or for consultation, which would reduce unnecessary patient traveling and the workload of urologists and other healthcare providers. We need to keep in mind, however, that patients do not only seek consultation for their oncological follow-up after RARP. It is prudent that the follow-up of patients after RARP also concerns urinary continence and erectile function. Luckily, today, patients can fill in digital patient-reported outcome measures (PROMS) in which data are reported on urinary function and/or erectile function and discuss these findings with their consulting urologist or nurse by telephone or in person if desired. Further treatment or advice can be given if indicated.

Further studies could focus on validating our findings. We encourage other authors to find a subgroup of low-risk patients who may benefit from a stripped-down and more personalized PSA follow-up scheme to reduce the burden on healthcare. Moreover, other studies could focus on evaluating the follow-up scheme of other treatment modalities for prostate cancer (i.e., external beam radiotherapy, brachytherapy, and systemic treatment) to reduce costs and healthcare burden. Lastly, it is important to find additional risk factors for a worse prognosis after RARP for a more tailored PSA follow-up scheme [20].

This study is inherently limited by its retrospective design. First, despite contacting referring centers in the PCNN periodically, 45% of our cohort did not have 1-year PSA follow-up and were, therefore, excluded from the analyses. It must be mentioned that 25% underwent surgery less than one year before the date of analysis. Second, our definition of BCP differed from the more commonly used definition of PSA >0.1 ng/mL 4–8 weeks after RP [13]. Therefore, our cohort may not fully correspond to other cohorts evaluating BCP. Due to our follow-up schedule, the first PSA test is performed 3–4 months after RARP. Therefore, we were unable to measure BCP after 5–8 weeks. Nonetheless, all patients who had BCP after 5–8 weeks also had PSA > 0.1 ng/mL after 4 months.

## 5. Conclusions

Patients with an unmeasurable PSA level at 3–4 months after RARP may not be retested until 12 months of follow-up, as the vast majority of patients will not have the biochemically recurrent disease at 5–8 months of follow-up. This will reduce PSA testing substantially at the cost of missing only a small number (1.2%) of patients with recurrent disease. With this, the burden on the healthcare system may be reduced. The efficacy of PSA testing could be even further improved by selecting patients for PSA measurements at different time points after RARP based on their prognostic pathological tumor characteristics. Patients with a higher risk of biochemical recurrence within one year after RARP have Gleason score ≥ 4 + 5 = 9, pathological seminal vesicle invasion, lymph node metastases, and positive surgical margins.

## Figures and Tables

**Figure 1 biomedicines-11-00727-f001:**
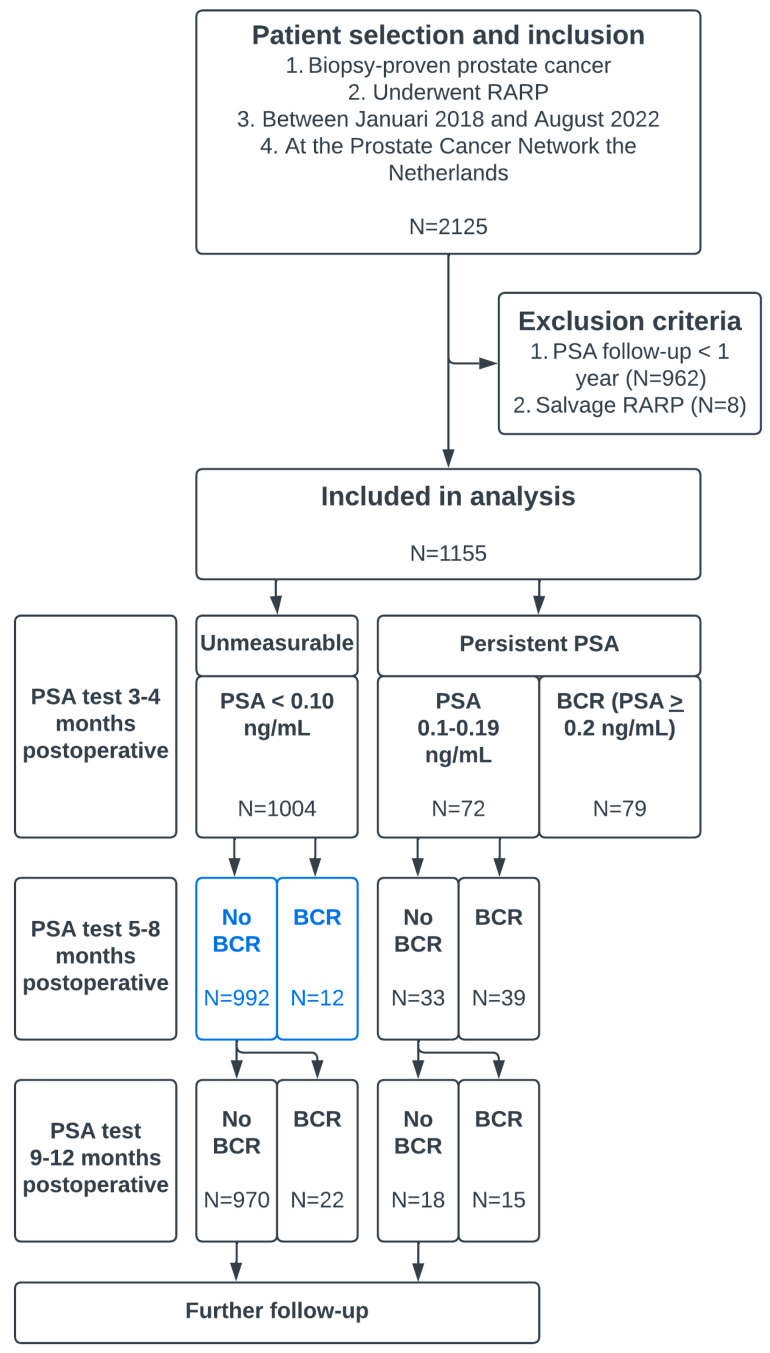
Flowchart describing patient selection and in- and exclusion criteria and a clarification of the progression of PSA testing and biochemical recurrence (BCR). The blue boxes represent patients who could be omitted from PSA testing 5–8 months after RARP.

**Figure 2 biomedicines-11-00727-f002:**
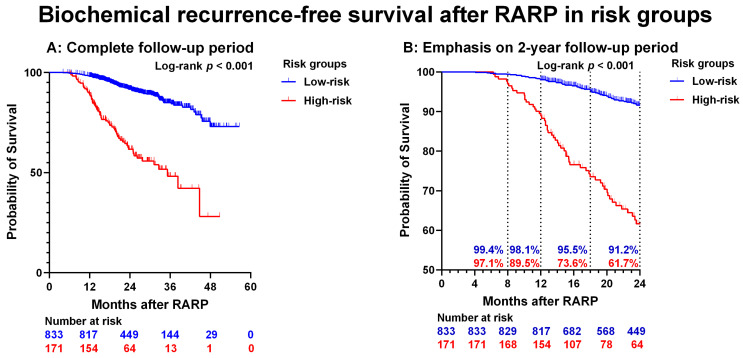
Biochemical recurrence-free survival after robot-assisted radical prostatectomy (RARP) in risk groups. (**A**): entire cohort. (**B**): Low- and high-risk only. The dotted lines represent the 8, 12, 18, and 24 months BCR-free survival rates. Please note that the Y-axis (probability of survival) shows values of 50–100, and the X-axis (Months after RARP) shows values of 0–24 months. Low-risk = absent BCP and both pathological ISUP grade 1–4 and pN0/pNx. High-risk = absent BCP and pathological ISUP grade 5 or pN1.

**Table 1 biomedicines-11-00727-t001:** Baseline characteristics of 1155 patients who underwent RARP at the Prostate Cancer Network in the Netherlands.

Total (N, %)	1155 (100%)
Age, years (median, IQR)	68 (63–72)
Pathological tumor stage (N, %)	
pT2	590 (51%)
pT3a	398 (35%)
pT3b	167 (15%)
Pathological nodal stage (N, %)	
pNx	526 (46%)
pN0	458 (40%)
pN1	171 (15%)
Pathological ISUP grade group (N, %)	
ISUP 1	66 (5.7%)
ISUP 2	540 (47%)
ISUP 3	328 (28%)
ISUP 4	55 (4.8%)
ISUP 5	162 (14%)
Unknown *	4 (0.4%)
Positive surgical margin (N, %)	414 (36%)
Biochemically persistent PSA (N, %)	151 (13%)
BCR ≤ 1 year after RARP (N, %)	168 (14%)
BCR after 0–4 months	79 (6.8%)
BCR after 5–8 months	51 (4.4%)
BCR after 9–12 months	37 (3.2%)

* In 4 patients, the pathologist was unable to determine the Gleason grade of the RARP specimen. BCR = biochemical recurrence, ISUP = international society of urological pathologists, RARP = robot-assisted radical prostatectomy.

**Table 2 biomedicines-11-00727-t002:** Crosstab between measurable PSA and BCR ≤ 1 year after RARP.

	Measurable PSA 3–4 Months Postoperative		
	No	PSA 0.1–0.199	PSA ≥ 0.2
BCR after 0–4 months	0	0	79 (100%)
BCR after 5–8 months	12 (1.2%)	39 (54%)	0
BCR after 8–12 months	22 (2.2%)	15 (21%)	0
No BCR or BCR after >1 year	970 (97%)	18 (25%)	0
Total	1004 (100%)	72 (100%)	79 (100%)

BCR = biochemical recurrence, RARP = robot-assisted radical prostatectomy.

**Table 3 biomedicines-11-00727-t003:** Logistic regression analysis for risk factors of BCR within 1 year after RARP in the subgroup of patients with unmeasurable PSA 3–4 m postop.

	Univariable		Multivariable 1		Multivariable 2	
	OR (95% CI)	*p*	OR (95% CI)	*p*	OR (95% CI)	*p*
pT2	ref	<0.001			ref	
pT3a	3.2 (1.4–7.7)	0.008			ref	
pT3b	8.4 (3.3–21)	<0.001			2.4 (1.0–5.5)	0.047
pNx	ref	<0.001	ref		ref	
pN0	1.8 (0.78–4.2)	0.17	ref		ref	
pN1	7.3 (2.9–18)	<0.001	3.6 (1.6–8.0)	0.02	3.1 (1.3–7.2)	0.008
ISUP 1–2	ref	<0.001	ref		ref	
ISUP 3	8.7 (2.9–26)	<0.001	ref		ref	
ISUP 4	6.6 (1.2–37)	0.032	ref		ref	
ISUP 5	18 (5.7–57)	<0.001	3.7 (1.7–8.0)	0.001	3.1 (1.4–7.0)	0.005
PSM	2.5 (1.3–5.0)	0.008	2.2 (1.1–4.4)	0.032		

As a rule of thumb, only 1 variable per 10 events was added to the multivariable logistic regression analysis. Since the outcome was present in 34 patients, only 3 variables at a time were added to the multivariable analysis. Therefore, two multivariable analyses were performed. BCR = biochemical recurrence, ISUP = international society of urological pathologists, pN-stage = pathological nodal stage, PSM = positive surgical margin, pT-stage = pathological tumor stage, RARP = robot-assisted radical prostatectomy, ref = reference categories.

## Data Availability

Data can be provided by the corresponding author upon reasonable request.

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
