# Peer review of "Identifying Patients in Whom the Follow-Up Scheme after Robot-Assisted Radical Prostatectomy Could Be Optimized in the First Year after Surgery: Reducing Healthcare Burden"

_biomedicines, 2023, doi:10.3390/biomedicines11030727_

Round 1

Reviewer 1 Report

The paper is absolutely very interesting and has the potential for application from practical point of view. The authors should address the following comments:

- The part of the introduction is very detailed and it is very important but in my opinion, it should not be in the introduction but in the introduction.

- The part of the abstract should be very concise, clear, and sharp, i.e., the abstract should contain the main results of the research and the methods for achieving the results. All the data that is present in the abstract can be moved to the introduction section.

- The introduction section is very short and not includes more relevant research in the considered fields.  

- The following paper can be relevant: 

1: Khan, S., Hicks, V., Rancilio, D., Langston, M., Richardson, K., & Drake, B. F. (2018). Predictors of Follow-Up Visits Post Radical Prostatectomy. In American Journal of Men’s Health (Vol. 12, Issue 4, pp. 760–765). SAGE Publications. https://doi.org/10.1177/1557988318762633

2: Labban, M., Dasgupta, P., Song, C., Becker, R., Li, Y., Kreaden, U. S., & Trinh, Q.-D. (2022). Cost-effectiveness of Robotic-Assisted Radical Prostatectomy for Localized Prostate Cancer in the UK. In JAMA Network Open (Vol. 5, Issue 4, p. e225740). American Medical Association (AMA). https://doi.org/10.1001/jamanetworkopen.2022.5740

3: Nave, O., & Elbaz, M. (2018). Method of directly defining the inverse mapping applied to prostate cancer immunotherapy — Mathematical model. In International Journal of Biomathematics (Vol. 11, Issue 05, p. 1850072). World Scientific Pub Co Pte Lt. https://doi.org/10.1142/s1793524518500729

- The analysis of the results presents very poorly. Please extended this section. 

- The conclusion section must be extended. 

- All the research that is present in this paper and only 2 plots?  

Good luck 

Author Response

Response to reviewer 1

The paper is absolutely very interesting and has the potential for application from practical point of view. The authors should address the following comments:

We thank the reviewer for their comment

1 The part of the introduction is very detailed and it is very important but in my opinion, it should not be in the introduction but in the introduction.

We thank the reviewer for their comment. We believe the reviewer suggests “in the introduction but in the discussion”? We believe the reader requires a general understanding of PSA testing after RARP and the costs/burden of the follow-up. Moving a large portion of the introduction to the discussion leads to misunderstanding our purpose. Also, 1. and 3. are conflicting statements. Therefore we decided not to change the introduction based on these comments.

2 The part of the abstract should be very concise, clear, and sharp, i.e., the abstract should contain the main results of the research and the methods for achieving the results. All the data that is present in the abstract can be moved to the introduction section.

We wholeheartedly agree with the first comment, but we respectfully disagree with the second comment. Moving the results part of the abstract to the introduction would make the abstract uninterpretable.

3 The introduction section is very short and not includes more relevant research in the considered fields.  

4 The following paper can be relevant: 

We thank the reviewer for these suggestions. We read the papers in order to assess the relevance for the current study.

  1. The aim of this study was to find predictors for follow-up (or loss to follow-up) after radical prostatectomy. The authors found that black men, unmarried men and men residing in a rural area have a higher chance of loss to follow-up. The aim of this study does not entirely match our study. Also, the Netherlands has a very dense population. One can travel from one end to the other in a 3-hours drive. Therefore, we have a maximum of 5% loss to follow-up for other reasons than death, severe illness, or re-referral to the GP or referring center. Loss to FU may be less of an issue in the Netherlands than in other less densely populations.
  2. The aim of this study is to evaluate the cost-effectiveness of RALP compared to LRP or ORP. They found that RALP has an incremental cost-effectiveness ratio of 4293 pounds/QALY and is more cost-effective than ORP or LRP, due to the lower BCR rate. All patients in our cohort were treated with RALP. Therefore, we adhere to the recommendation to use ralp.
  3. We do not believe the third paper has any relation with the current study.

1: Khan, S., Hicks, V., Rancilio, D., Langston, M., Richardson, K., & Drake, B. F. (2018). Predictors of Follow-Up Visits Post Radical Prostatectomy. In American Journal of Men’s Health (Vol. 12, Issue 4, pp. 760–765). SAGE Publications. https://doi.org/10.1177/1557988318762633

2: Labban, M., Dasgupta, P., Song, C., Becker, R., Li, Y., Kreaden, U. S., & Trinh, Q.-D. (2022). Cost-effectiveness of Robotic-Assisted Radical Prostatectomy for Localized Prostate Cancer in the UK. In JAMA Network Open (Vol. 5, Issue 4, p. e225740). American Medical Association (AMA). https://doi.org/10.1001/jamanetworkopen.2022.5740

3: Nave, O., & Elbaz, M. (2018). Method of directly defining the inverse mapping applied to prostate cancer immunotherapy — Mathematical model. In International Journal of Biomathematics (Vol. 11, Issue 05, p. 1850072). World Scientific Pub Co Pte Lt. https://doi.org/10.1142/s1793524518500729

5 The analysis of the results presents very poorly. Please extended this section. 

We do not fully understand this comment. We request the reviewer to please elaborate on which part of the results requires improved presentation and we will alter it.

6 The conclusion section must be extended. 

We added the following sentence to the conclusion section: Patients with a higher risk of biochemical recurrence within one year after RARP have Gleason score > 4 + 5 = 9, pathological seminal vesicle invasion, lymph node metastases and positive surgical margins.

7 All the research that is present in this paper and only 2 plots?  

Please do not overlook the 3 tables. It is possible to add more Kaplan-Meier curves of the risk factors shown in table 3, but we already know from a vast body of literature that these variables are risk factors for BCR. Moreover, we believe that adding these plots distracts the reader from our main message being that 99% of PSA tests can be safely omitted at 5-8 months after RARP.

Good luck 

Reviewer 2 Report

This article evaluates the follow-up scheme of patients undergoing RARP for locally-confined PCa could be optimized.

The currently advised follow-up scheme of PSA-testing after robot-assisted radical prostatectomy (RARP) is strict and might pose a burden to our healthcare system.

The goal set by the authors is to try to optimize the 1-year follow-up scheme for patients undergoing RARP.They evaluated patients who had undergone RARP between 2018 and 2022, have collected post-operative PSA values, defining biochemical persistence as PSA values >0.10 ng/mL at 0-4 months after RARP, while biochemical recurrence (BCR) was defined as 

PSA level >0.2 ng/mL at any time after RARP.

After these assessments, they found that patients who presented with an unmeasurable PSA value 3-4 months after RARP, may not be retested until 12 months as about 100% of patients will not have a biochemical recurrence at 5-8 months follow-up, this will substantially reduce the PSA test.

Despite this, the article has numerous shortcomings, focuses only on possible cost reduction in terms of follow-up in patients with localized prostate cancer who have undergone rarp.

It would be interesting to also broaden the focus on locally advanced prostate carcinomas and what the possible cost-benefit impacts might be, or even go to evaluate for that type of tumor what might be new types of therapeutic.

Other remarkably interesting insights might consider hormone therapy who could play increasingly important roles.

1.     Regarding the topics covered, it would be interesting to take a cue from this article 10.3390/diagnostics11050908,as it could give new insights to the article and enrich it with potential. 

Author Response

Reviewer 2

This article evaluates the follow-up scheme of patients undergoing RARP for locally-confined PCa could be optimized.

The currently advised follow-up scheme of PSA-testing after robot-assisted radical prostatectomy (RARP) is strict and might pose a burden to our healthcare system.

The goal set by the authors is to try to optimize the 1-year follow-up scheme for patients undergoing RARP.They evaluated patients who had undergone RARP between 2018 and 2022, have collected post-operative PSA values, defining biochemical persistence as PSA values >0.10 ng/mL at 0-4 months after RARP, while biochemical recurrence (BCR) was defined as 

PSA level >0.2 ng/mL at any time after RARP.

After these assessments, they found that patients who presented with an unmeasurable PSA value 3-4 months after RARP, may not be retested until 12 months as about 100% of patients will not have a biochemical recurrence at 5-8 months follow-up, this will substantially reduce the PSA test.

Despite this, the article has numerous shortcomings, focuses only on possible cost reduction in terms of follow-up in patients with localized prostate cancer who have undergone rarp.

It would be interesting to also broaden the focus on locally advanced prostate carcinomas and what the possible cost-benefit impacts might be, or even go to evaluate for that type of tumor what might be new types of therapeutic.

Other remarkably interesting insights might consider hormone therapy who could play increasingly important roles.

We commend the reviewer for their thorough analysis and complete understanding of the paper. We agree that it is interesting to broaden the focus to more advanced prostate cancer, other types of treatment and the role of hormone therapy in cost-reduction. We believe that a similar study should be performed for, for example, external beam radiotherapy for locally confined prostate cancer. However, we focused on the treatment of locally confined prostate cancer with RARP. We collect data from all patients who underwent RARP in our hospital network, therefore we have an extensive database with limited bias. We do not routinely collect data for other types of treatment and for advance prostate carcinoma, therefore, we cannot answer the questions proposed by the reviewer. However, we will include their suggestion in the ‘future perspectives’  section of the discussion.

  1. Regarding the topics covered, it would be interesting to take a cue from this article 10.3390/diagnostics11050908,as it could give new insights to the article and enrich it with potential. 

We agree that it is important to find factors that are associated with a worse prognosis after curative treatment for prostate cancer, as we tried to show in the current paper. BRCA germline mutation seems to be such a risk factor. We believe it is important to find additional risk factors for a more tailored follow-up scheme.

We added the following sentence to the ‘future perspectives’ section of the discussion.

Lastly, it is important to find additional risk factors for worse prognosis after RARP for a more tailored PSA follow-up scheme[20].

Reviewer 3 Report

The manuscript by Veerman et.al. explored the possibility to optimize post op follow up of RALP by omitting a PSA check. Authors identified risk factors for BCR and reached the conclusion that  for certain patients it is safe to omit one PSA check which can potentially reduce health care burden. Some comments

1. Table 3, please provide more information on why two multivariable analyses were performed

2. Figure 2. Panel A was labelled as "entire cohort" and it contained the same two curves of low and high risk as shown in Panel B. It looks like both panels only differed by the time on x axis. Please clarify the difference between the two panels

Author Response

Reviewer 3

The manuscript by Veerman et.al. explored the possibility to optimize post op follow up of RALP by omitting a PSA check. Authors identified risk factors for BCR and reached the conclusion that  for certain patients it is safe to omit one PSA check which can potentially reduce health care burden. Some comments

  1. Table 3, please provide more information on why two multivariable analyses were performed

Below the table we explained why two multivariable analyses were performed:  “As a rule of thumb, only 1 variable per 10 events was added to the multivariable logistic regression analysis. Since the outcome was present in 34 patients, only 3 variables at a time were added to the multivariable analysis.”

We added: “Therefore, two multivariable analyses were performed.”

  1. Figure 2. Panel A was labelled as "entire cohort" and it contained the same two curves of low and high risk as shown in Panel B. It looks like both panels only differed by the time on x axis. Please clarify the difference between the two panels

We thank the reviewer for noticing this inconsistency. In an earlier version, patients with biochemical persistent PSA were also included in panel A as a third group. We decided to remove them to emphasize the risk groups. However, we forgot to change the titles. We created a new version with correct titles.

Round 2

Reviewer 1 Report

The authors revised the paper according to my comments. One issue should be solved before publication: Please edit the paper in English mother language. 

Reviewer 2 Report

Authors answered all comments and suggestions.